# The Association of uPA, uPAR, and suPAR System with Inflammation and Joint Damage in Rheumatoid Arthritis: suPAR as a Biomarker in the Light of a Personalized Medicine Perspective

**DOI:** 10.3390/jpm12121984

**Published:** 2022-12-01

**Authors:** Maurizio Benucci, Arianna Damiani, Edda Russo, Serena Guiducci, Francesca Li Gobbi, Paola Fusi, Valentina Grossi, Amedeo Amedei, Mariangela Manfredi, Maria Infantino

**Affiliations:** 1Rheumatology Unit, Hospital S. Giovanni di Dio, Azienda USL-Toscana Centro, 50143 Florence, Italy; 2Department of Clinical and Experimental Medicine, University of Florence, 50134 Florence, Italy; 3Immunology and Allergology Laboratory, Hospital S. Giovanni di Dio, Azienda USL-Toscana Centro, 50143 Florence, Italy

**Keywords:** suPAR, uPA/uPAR, rheumatoid arthritis, disease activity score, systemic chronic inflammation

## Abstract

Background: In recent years, the involvement of the soluble urokinase Plasminogen Activator Receptor (suPAR) in the pathophysiological modulation of Rheumatoid Arthritis (RA) has been documented, resulting in the activation of several intracellular inflammatory pathways. Methods: We investigated the correlation of urokinase Plasminogen Activator (uPA)/urokinase Plasminogen Activator Receptor (uPAR) expression and suPAR with inflammation and joint damage in RA, evaluating their potential role in a precision medicine context. Results: Currently, suPAR has been shown to be a potential biomarker for the monitoring of Systemic Chronic Inflammation (SCI) and COVID-19. However, the effects due to suPAR interaction in immune cells are also involved in both RA onset and progression. To date, the literature data on suPAR in RA endorse its potential application as a biomarker of inflammation and subsequent joint damage. Conclusion: Available evidence about suPAR utility in the RA field is promising, and future research should further investigate its use in clinical practice, resulting in a big step forward for precision medicine. As it is elevated in different types of inflammation, suPAR could potentially work as an adjunctive tool for the screening of RA patients. In addition, a suPAR system has been shown to be involved in RA pathogenesis, so new data about the therapeutic response to Jak inhibitors can represent a possible way to develop further studies.

## 1. Introduction

Rheumatoid arthritis (RA) is a chronic autoimmune inflammatory disease that affects the physiology of multiple joints due to abnormal bone erosion and cartilage deterioration caused by a variety of elements, such as the matrix metalloproteinase (MMP) and a disintegrin and metalloproteinase with thrombospondin motifs (ADAMTS) produced by chondrocytes, synovial fibroblasts, and synovial macrophages [1,2]. This process involves cells from the myeloid and leukocyte lineages, including monocytes/macrophages, neutrophils, mast cells, B lymphocytes, and some subsets of T helper cells [3,4]. Moreover, cytokines in the joint space, including tumor necrosis factor-alpha (TNF), interleukin 1 beta (IL-1β), IL-6, and IL-17, contribute to the pleiotropic damage of the joints in RA [5]. Recently, other emerging elements have been documented to be involved in the pathophysiological modulation of RA. In particular, there is the plasminogen activator (PA) complex, a family of proteins that has been identified to carry out this pleiotropic process [6]. Of note, plasminogen is an abundant plasma protein that exists in various zymogenic forms. Plasmin, the proteolytically active form of plasminogen, is known for its essential role in the breakdown of blood clots (fibrinolysis). In detail, the PA system is an extracellular proteolytic enzyme structure divided into two forms, tissue-type PA (tPA) and urokinase PA (uPA), which are expressed during a variety of cellular activities [7]. In particular, the uPA family of proteins is the major constituent of the PA system, which is involved in inflammatory processes, including tissue remodeling as well as tumor progression and metastasis. These proteins are recognized by the uPA receptor (uPAR), which activates diverse intracellular inflammatory pathways [8]. Indeed, uPA–uPAR binding promotes the change of plasminogen to plasmin, which in turn stimulates a succession of proteolytic cascades to destroy extracellular matrix elements. Furthermore, interaction with coreceptors such as integrins and vitronectin promotes recognition (uPAR ligands) [9]. However, in normal physiological conditions, uPA fulfills its function in the tPA presence [10]. In recent years, uPA protease has been strongly implicated in the pathogenetic process and progression of cartilage damage in RA. This physiological process regulates several cellular pathways, including cytokine secretion, cell activation/migration, and fibrinolysis [11,12]. All of these processes begin with an interaction between uPA and its receptor uPAR, which causes tissue remodeling and T cell activation [13]. Moreover, increased uPA expression and decreased tPA expression have been related to the severity of RA disease [14]. In addition, the activity of synovial cells such as macrophages, fibroblast-like synoviocytes (FLS), chondrocytes, and endothelial cells is regulated by the uPA/uPAR interaction, allowing them to secrete a variety of cytokines, chemokines, and growth factors that alter the prognosis of RA [15]. In the absence of macrophage colony-stimulating factor (M-CSF), uPA/uPAR expression suppresses osteoclast differentiation/formation via upregulation of adenosine monophosphate-activated protein kinase (AMPK) [16]. Conversely, other data have demonstrated that in the presence of M-CSF, uPAR promotes osteoclast differentiation via a PI3K/Akt-dependent mechanism [17]. Moreover, other transcription factors (TFs) that it can activate include nuclear factor kappa B (NFB) and nuclear factor activator of T cells 1 (Nfatc1) [17,18].

In addition, there is a soluble form of uPAR (suPAR) that results from the cleavage of membrane-bound uPAR during immune activation or inflammation. suPAR is found in a variety of biological fluids, including urine, plasma, cerebrospinal fluid, blood, and serum, and its concentration is related to immune system activation [19].

From this perspective, based on the increasing application of suPAR as a biomarker for the monitoring of Systemic Chronic Inflammation (SCI) [19], we examined the effects due to uPA/uPAR interaction in the immune cells involved in RA onset and progression. Then, we further discuss the recent findings on suPAR in RA, exploring its future potential application as a biomarker of inflammation and progressive joint damage.

## 2. Materials and Methods

The literature on existing evidence on uPA/uPAR expression and suPAR in RA has been revised, structuring the manuscript as a perspective. The research was carried out using PubMed, Scopus, and EMBASE databases, searching for the following key words: uPA/uPAR, suPAR, rheumatoid arthritis, Systemic Chronic Inflammation, disease activity score, and biomarkers. Only English language articles have been selected. Additional references were identified by a manual search among the cited references. Abstracts without a main text have been excluded from the perspective.

## 3. Results

### 3.1. Molecular Biochemical Aspects of suPAR

From the molecular point of view, suPAR is derived from the cleavage and release of the membrane-bound protein uPAR expressed by immune cells, trophoblasts, endothelial cells, smooth muscle cells, and certain tumor cells [20,21]. uPAR is a GPI-linked receptor expressed by immune cells upregulated at sites of inflammation and tissue remodeling [22,23]; in addition, it is involved in the fibrinolysis cascade [24]. Active uPA can, in fact, cleave plasminogen to plasmin, which then activates uPA. Plasmin stimulates cell migration and invasion, fibrinolysis, vasodilation, opsonization, and the phagocytosis of foreign pathogens. In addition, it is able to activate matrix metalloproteases (MMPs), cleave extracellular matrix (ECM) components, degrade fibrin, and activate the classical complement pathway. In detail, uPAR interacts with a variety of ligands and receptors, a large percentage of which are integrins, to promote intracellular signaling, cell migration, cell adhesion, and tissue remodeling [7]. The protein is composed of three domains, D1–D3, linked with a linking region between D1 and D2–D3 (Figure 1). The cleavage sites are found in the linker region and in the GPI anchor, and the three major suPAR isoforms are full-length suPAR I–III, suPAR I, and suPAR II–III. Interestingly, cleavage of uPAR/suPAR in the linker region exposes an SRSRY sequence, which is involved in chemotaxis. Finally, suPAR is detectable in low, but constant, concentrations in the plasma of healthy individuals [25,26].

### 3.2. suPAR as a Novel Biomarker for Systemic Chronic Inflammation (SCI) and COVID-19

SCI is a chronic, health-damaging, low-grade inflammation that has a pivotal function in immunosenescence (the age-related decline in immune function) and in the development and progression of many diseases such as cardiovascular disease, cancer, diabetes mellitus, chronic kidney disease, non-alcoholic fatty liver disease, and autoimmune and neurodegenerative disorders. The clinical SCI effects are linked to several disorders across organ systems and include increased risk of physical frailty, morbidity, and mortality in the previously mentioned chronic and acute diseases. Since there are presently no recognized and accepted biomarkers for assessing SCI levels alone, SCI is usually measured by combining the clinical biomarkers of acute inflammation and infection, such as CRP, IL-6, and TNF-α [27].

However, recently Rasmussen et al. described at least 10 properties and characteristics shared by suPAR and SCI [19], which can be evaluated to support the potential role of suPAR as a biomarker. Indeed, SuPAR has the potential to be a robust biomarker and the new gold standard for measuring SCI. In particular, (1) immune activation enhances suPAR expression and release; (2) uPAR and suPAR have pro-inflammatory activities; (3) suPAR is linked to the amount of circulating immune cells; (4) suPAR correlates with the levels of currently used inflammatory biomarkers; and (5) suPAR is unaffected by acute and short-term changes, unlike many currently used markers of systemic inflammation. (6) SuPAR, like SCI, has been linked to a wide range of diseases. (7) SuPAR and SCI both indicate morbidity and mortality. (8) SuPAR and SCI have risk factors in common. (9) SuPAR, along with other inflammatory biomarkers, is associated with inflammation risk factors and outcomes. (10) SuPAR levels can be reduced by anti-inflammatory treatments and therapeutic applications.

As a result, defining the SCI level is critical because it can provide information on the disease burden as well as the risk of disease onset, progression, and, ultimately, mortality. In this context, blood suPAR monitoring may be a newer potential marker of SCI because it is stably connected with inflammation and immune activation and shares risk factors with many age-related diseases. There is also strong proof that suPAR can be used to predict adverse events, morbidity, and mortality. It has been linked to immune activity and prognosis in a number of diseases, including RA.

Moreover, due to its correlation with inflammation, suPAR has recently already been evaluated as a biomarker during SARS-CoV-2 infection [28,29]. Indeed, the early rise in the serum levels of suPAR is associated with increased risk of coronavirus disease 2019 (COVID-19) progression to respiratory failure [29].

The “SAVE-MORE” study evaluated the safety and efficacy of Anakinra, an IL-1α/β inhibitor, in a large cohort of COVID-19 patients at risk of progression to respiratory failure, with a level of suPAR ≥ 6 ng/mL. On day 28, Anakinra resulted in a median decrease in the Sequential Organ Failure Assessment (SOFA) score, reduced mortality, and a shorter hospital stay [30]. IL-1α is also a cytokine implicated in RA, and Anakinra was also initially studied in the treatment of RA [31].

Regarding the diagnostic performance of the marker, a recent study investigated the value of suPAR as a prognostic tool in comparison with other variables regarding disease severity and length of hospital stay in patients with COVID-19 compared to healthy controls [32]. SuPAR, blood cell counts, lactate dehydrogenase (LDH), C-reactive protein (CRP), estimated glomerular filtration rates, and plasma creatinine were measured at admission, and level of care, oxygen demand, and length of stay were recorded. The SuPAR levels in patients were significantly higher than in controls. The levels were greater in severe/critically ill patients than in moderately ill patients. In addition, suPAR levels correlated with length of hospitalization. Besides suPAR, LDH, CRP, neutrophil count, neutrophil-to-monocyte and neutrophil-to-lymphocyte ratio, body mass index and chronic renal failure were discriminators of COVID-19 severity and/or predictors of length of hospitalization. This study showed that suPAR functioned as an independent predictor of COVID-19 disease severity [32].

### 3.3. uPA/uPAR Secretion by Immune Cells in Rheumatoid Arthritis

Several immune cell types derived from the myeloid and leukocyte lineages stimulate pleiotropic joint physiology in the RA microenvironment. As previously stated, uPA/uPAR signaling is thought to have a pivotal role in modifying the disease condition of RA. Indeed, during RA disease, cell phenotypes such as articular chondrocytes, neutrophils, and monocytes secrete large amounts of uPA (Figure 2).

The Table 1 shows the immune cell types producing uPA, the pathways activated by uPA/uPAR signaling, and the final effect on RA.

#### 3.3.1. uPA/uPAR System in Chondrocytes

The inflammatory microenvironment, via cytokines such as TNF, IL-1, and IL-6 as well as growth factors such as platelet-derived growth factors (PDGF) and vascular endothelial growth factor (VEGF), stimulates chondrocytes to secrete different classes of matrix metalloproteinases (MMPs) that disrupt the extracellular matrix (ECM), thereby modifying bone morphology [37,38]. In RA, chondrocytes play a key role in altering the cartilage and bone morphology, causing numerous deformities and erosions [39]. In RA, chondrocytes that secrete/express uPA and its receptor are being studied for their ability to influence immunomodulatory changes in joint disease [40]. Chondrocytes effectively express uPAR, which recognizes the cleavage form of uPA circulating in the synovium and enhances the gradual destruction of the joint region via TNF, IL-1, and retinoids secretion [41]. In addition, VEGF promotes integrin-mediated β1 expression of uPAR in chondrocytes and triggers MMP secretion, which induces progressive destruction of joint cartilage [34]. In the RA synovium, pro-inflammatory cytokines from myeloid cells/leukocytes actively regulate the levels of uPA secreted by chondrocytes [10]. The interplay of uPA/uPAR expression in chondrocytes promotes pericellular proteolysis, which is a crucial component in cartilage degradation [42]. Furthermore, IL-1 activation of articular chondrocytes triggers the production of HIF-1, plasminogen activator inhibitor 1 (PAI-1), and HIF 1 silencing reduced PAI-1 levels [43].

#### 3.3.2. uPA/uPAR System in Neutrophils

Neutrophils are the most common polymorphonuclear population leukocytes (PMNs) in the joint, with the ability to release cytotoxic degradative enzymes and generate reactive oxygen species (ROS) [44]. They also release a wide range of cytokines and chemokines that affect the responses of other immune cells via cell-to-cell contact. [45]. Furthermore, Neutrophil extracellular traps (NETs) are released in large quantities by neutrophils and promote self-antigen reactions, resulting in humoral immune responses [46]. This is a source of the autoantigens that promote an autoimmune condition like RA [47,48]. Neutrophils also release high amounts of uPA into the synovial fluid, which starts some inflammatory processes [35]. A strong urokinase activity has been found in the synovial fluid of RA [49]. A correlation was documented between uPA levels and neutrophil counts with arthritogenic properties [50].

#### 3.3.3. uPA/uPAR System in Monocytes/Macrophages

Monocytes/macrophages are the most crucial antigen-presenting cells (APCs), secreting a variety of cytokines (TNF, IL-6, IL-1, and IL-8), growth factors, and chemokines, which influence RA pathophysiology (MCP1, VEGF, GM-CSF, and MIP-1) [51,52]. They activate tissue macrophages resident in the synovia and promote the differentiation and proliferation of T cells and B cells [53,54]. Furthermore, monocytes stimulated by RANKL and M-CSF undergo unregulated differentiation into osteoclasts, resulting in abnormal bone erosion [55,56].

In addition to these well-established features, during the progression of RA disease, monocytes/macrophages actively produce and/or express the uPA/uPAR system to start and mediate the inflammatory process [36]. Moreover, macrophages were found to express large amounts of suPAR [57]. Furthermore, increased suPAR expression in RA patients’ synovial macrophages has been shown to enhance fibrin clot formation and the inflammatory cytokine pattern, giving rise to immune cell infiltration and modification of joint pathophysiology [58]. Neutrophil suPAR has been observed to increase inflammation and macrophage proliferation [59]. Moreover, suPAR released by neutrophils increases macrophage phagocytosis and cytokine release during RA progression [60]. Other factors such as macrophage migration inhibitory factor (MIF) and VEGF are also co-expressed with uPAR and are attenuated by adalimumab treatment [61]. These processes are halted by the addition of a specific uPA inhibitor, such as (PAI-1) [62].

#### 3.3.4. uPA/uPAR System in Fibroblast-Like Synoviocytes (FLS)

Type B synoviocytes, also known as FLS, are the most common type of cell found in the synovial lining [63]. FLS proliferate in a similar way to that of a cancerous condition during the progression of RA and enhance different immunomodulatory processes [64]. FLS produce and release a diverse range of inflammatory cytokines (TNF, IL-6, IL-1, and IL-23), chemokines (IL-8 and MCP-1), and effector molecules (M –CSF, RANKL, and MMP) [65,66].

An initial study found that RA synovial fibroblasts (RASF) expressed high levels of uPA rather than tPA, leading to aggressive tumor-like behavior [67].

Many reports have revealed that periarticular fibrinolytic activity in RASF was increased by the uPA/uPAR interplay and inhibited by hyaluronic acid (HA) treatment [68,69]. Furthermore, through aggressive fibrinolytic behavior, the uPA system increased the survival/proliferation of RA FLS cells [33]. The cells’ interaction with the uPA/uPAR system expressed in fibroblasts triggered tumorigenic activity via tyrosine, PI3K, and mitogen kinase activity [70]. According to new research, uPAR expression in RA-FLS cells promotes aggressive tumor-like propagation by inducing the 1/PI3K/Akt integrin signaling pathway and reduces neoangiogenesis in RA patients [71].

#### 3.3.5. uPA/uPAR System in Endothelial Cells

Endothelial cells of the vessels are present in the lining of the synovium and are involved directly in inflammatory disease processes such as RA. [72]. TNF and IL-1 are synovial cytokines that enhance the expression of VCAM-1, ICAM-1, and E-selectin on endothelial cells [73]. The expression of these cell adhesion molecules fosters the migration of different leukocyte populations to the joint area, causing a reduction in synovial vascularity [67].

The expression of these cell adhesion molecules promotes the migration of various leukocyte populations to the joint, resulting in a decrease in synovial vascularity [74].

### 3.4. Signaling Pathways Mediated through uPA/uPAR during RA Progression

uPA, which is secreted by myeloid/leukocyte cell types, interacts with its receptor uPAR, which is expressed on macrophages and FLS, to activate a variety of inflammatory responses. The interplay of uPA/uPAR and the induction/attenuation of several effectors mediates the inflammatory process and its progression.

Regarding RA, AMPK activation inhibits several inflammatory pathways, which include Janus kinase/signal transducer and transcription activator (JAK/STAT), PI3K/Akt, sirtuin 1 (SIRT1), NFB, and the cAMP response element binding protein pathways (CREB) [75]. Numerous inflammatory factors found in the synovium suppress AMPK induction within cells [76]. In the RA, the uPA/uPAR signaling cascade is critical for regulating AMPK expression.

AMPK activation in joint chondrocytes is known to reduce the uPA released into the synovial joint space [40]. In contrast, the uPA/uPAR interaction has been shown to promote the inhibition of NFκB-dependent osteoclastogenesis through activation of the plasmin/PAR-1/Ca^2+^/CaMKK/AMPK axis [18]. In RA, the PI3K/Akt signaling pathway mediates the induction of a cancer-like condition that results in the aberrant/uncontrolled growth and differentiation of multiple cellular phenotypes in the synovium [77]. These processes are mediated by PI3K by activating protein kinase B (PKB/Akt) and other downstream components such as the mechanistic target of rapamycin (mTOR), CREB, and nuclear factor kappa B (NFB) [78]. Furthermore, in a co-culture system with osteoblasts, mesenchymal stem cells (MSCs) differentiate into mature osteoclastic cells in the absence of uPAR expression [79]. Osteoblasts secreting M-CSF in conjunction with uPAR increased osteoclastogenesis via interaction with cFMS and cell survival via induction of PI3K/Akt/NFB signaling [79]. Similarly, multiple studies have demonstrated that uPAR inhibits LPS/M-CSF-induced osteoclastogenesis by inhibiting the integrin/Akt/NFB signaling pathways [16,18]. uPAR promotes tumor-like pleomorphic changes in RA-FLS cells isolated from patients by upregulating 1-integrin/PI3K/Akt signaling [71]. The interaction of these two pathways emphasizes the importance of uPA/uPAR in osteoclastogenesis and bone remodeling in RA. As a whole, these findings indicate that uPAR expression plays a dual role in the presence/absence of growth factors such as M-CSF, dampening osteoclast formation in a disease condition such as RA.

### 3.5. suPAR as a New Potential Biomarker in Rheumatoid Arthritis

The activation of the immune system and the development of an inflammatory response leads to elevated plasma suPAR concentrations. As previously mentioned, suPAR is a biomarker increasingly used for the monitoring of SCI.

A few previous studies have investigated circulating suPAR in RA [36,61] documenting that their levels were increased in RA patients compared to healthy controls [36,61]. In addition, suPAR levels correlated with the number of swollen joints [36,80], also among patients with limited disease activity [80]. However, none of the previously published studies addressed early RA, i.e., a phase of particular importance to tailor anti-rheumatic treatment to prevent future joint damage and disability [81]. In a recent study, serum suPAR was evaluated by an enzymatic immunosorbent test at disease onset and after 3 and 36 months, in 252 patients from a Swedish prospective observational cohort with early RA [82]. The suPAR levels and changes in relation to the joint disease activity score at 28 (DAS28) and joint damage according to the Larsen score at inclusion and during follow-up were analyzed. In addition, 100 healthy blood donors were also observed as controls. The suPAR circulating levels were always higher in RA point patients than in healthy controls. Baseline suPAR was significantly associated with underlying disease activity, while suPAR levels at 36 months were associated with joint damage at 36 months. No predictive values of suPAR’s levels or variations were found over time. The authors concluded that suPAR levels were associated with disease activity in early untreated RA and reflect joint damage in later stages. Increased suPAR in established RA could indicate patients needing frequent joint monitoring, regardless of disease activity. From the perspective of suPAR as a rapidly rising biomarker, it is relevant to be aware of its ability to reflect both inflammation and subsequent joint damage [82].

## 4. Perspectives and Research Agenda

The research on RA pathophysiology indicates how the uPA/uPAR system could be linked to the different cellular components of the disease. The presence of progressive inflammation and cartilage and bone damage suggests that this system affects the stages of the pathogenesis. In light of these observations, the persistence of circulating levels of suPAR may represent a promising new biomarker of RA progression. In addition, suPAR might also act as a predictive biomarker of response to biologic disease-modifying antirheumatic drug (b-DMARD) and targeted synthetic disease-modifying antirheumatic drug (ts-DMARD) therapies). Indeed, a decrease in good responders to Adalimumab was observed, while these changes were not observed in non-responders [61]. The elevated circulating levels at baseline and persistent over time in non-responders versus persistent good responders could also identify the population of difficult-to-treat (D2T) RA patients requiring Jak inhibitor therapies, which interfere with multiple cytokine pathways of inflammation as well as phases of innate immunity, adaptive immunity, and antibody production by B lymphocytes.

## 5. Conclusions

In the field of RA precision medicine, the utility of suPAR is promising, so future research should further explore its application in clinical practice. Indeed, from this perspective, we suggest that suPAR detection may be developed as an adjunctive tool for the screening of RA patients, but larger cohort studies that support these findings must be performed. Moreover, the ability of Jak inhibitors to interfere with intracellular systems via the JAK/STAT pathway may represent a ground for future research. Data on circulating suPAR levels, which are correlated with disease activity and are a response to treatment with Jak inhibitors, could identify a possible predictor for a group of patients on which to target such treatments for personalized medicine for rheumatoid arthritis.

## Figures and Tables

**Figure 1 jpm-12-01984-f001:**
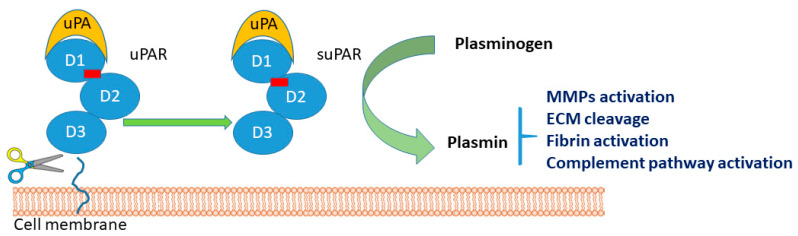
Schematic representation of the urokinase plasminogen activation (uPA) system. D1, D2, and D3 represents the three domains of the protein. The red line represents the linker region between D1 and D2–D3 domains. Active uPA cleaves plasminogen to plasmin that activates matrix metalloproteases (MMPs degrade fibrin), cleaves extracellular matrix (ECM) components, and activates the complement pathway.

**Figure 2 jpm-12-01984-f002:**
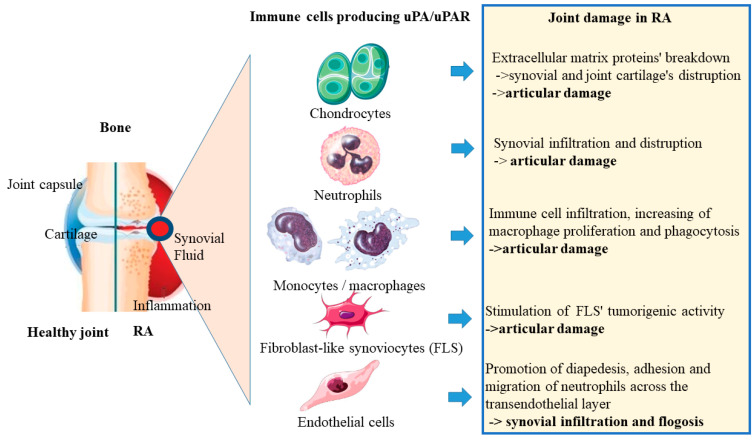
Immune cells producing uPA/uPAR as articular chondrocytes, neutrophils, monocytes, FLS, and endothelial cells produce high amounts of uPA in the RA microenvironment during RA disease condition, promoting articular damage, synovial infiltration, and phlogosis in the joint.

**Table 1 jpm-12-01984-t001:** Cell types, pathways activated, and effect of uPA/uPAR signaling on Rheumatoid Arthritis [33,34,35,36].

Cell Type	Pathways Activated by uPA/uPAR	Effect on RA
Chondrocytes	MMP1, MMP3 and MMP13, TNF, IL-1β, and retinoids secretion	Extracellular matrix proteins’ breakdown -> synovial and joint cartilage’s distruption -> articular damage
Neutrophils	Stimulation various of inflammatory processes (releasing of cytotoxic degradative enzymes, cytokines, chemokynes, NETs formation)	Synovial infiltration and distruption -> articular damage
Monocytes/macrophages	Improvment of fibrin clot formation, cytokine release	Immune cell infiltration, increasing of macrophage proliferation and phagocytosis -> articular damage
Fibroblast-like synoviocytes (FLS)	Promotion of fibrinolytic activity, stimulation of β1/PI3K/Akt integrin, tyrosine and mitogen kinase’s signaling pathways	Stimulation of FLS’ tumorigenic activity -> articular damage
Endothelial cells	Proteolytic activity	Promotion of diapedesis, adhesion and migration of neutrophils across the transendothelial layer -> synovial infiltration and flogosis

## Data Availability

Not applicable.

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
