# Peer review of "The Association of uPA, uPAR, and suPAR System with Inflammation and Joint Damage in Rheumatoid Arthritis: suPAR as a Biomarker in the Light of a Personalized Medicine Perspective"

_jpm, 2022, doi:10.3390/jpm12121984_

Round 1

Reviewer 1 Report

Dear Author,

This overview is very interesting and it show that new sensitive tool biomarker could be available for RA patients as an important treatment target.

Well written English, valuable for publishing.

Best regards,

Dragana Lazarevic

Author Response

We thank the reviewer for her positive comment regarding our perspective.

Best regards

Reviewer 2 Report

The research is focused on the validation of another biomarker in rheumatoid arthritis. suPAR participates in the pathophysiological modulation of rheumatoid arthritis, activating intracellular inflammatory pathways. It also participates in the onset of the disease itself and negatively affects its development in terms of progression. It has a similar effect in other chronic inflammatory diseases as well as in Covid-19 infection. It is believed that in a therapeutic sense it can act as a JAK inhibitor and thus reduce joint damage. Long-term researches stimulated by this work are expected.

Author Response

We thank the reviewer for her positive comment regarding our perspective. 

Reviewer 3 Report

This manuscript was written in the form of a systematic review. However, this paper is not organized in the basic format that a systematic review should have. In addition, the content of the results is insufficient to support the conclusions claimed in this paper.

I reject this paper because it lacks a scientific method and it is judged that the basis for its conclusion is insufficient.

Author Response

We thank the reviewer for his/her comment. However, our review is not organized in the basic format of a systematic review because our intent was to present literature data in the form of a perspective review, as reported in the tile and in the Materials and Methods section. We chose this format following the previous editor’s suggestion after our first submission. Indeed, in our perspective review it is not present any identification of bias.

We followed the general rules for writing a perspective,  as reported below.

Round 2

Reviewer 3 Report

I fully understood the author's response, and reviewed the format of the paper again. And, I think this manuscript is good enough to be published.